# A Conceptual Model Framework for Mapping, Analyzing and Managing Supply–Demand Mismatches of Ecosystem Services in Agricultural Landscapes

**Mostafa Shaaban \*, Carmen Schwartz, Joseph Macpherson and Annette Piorr** 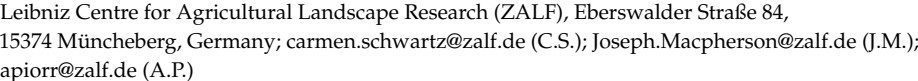

Leibniz Centre for Agricultural Landscape Research (ZALF), Eberswalder Straße 84,
15374 Müncheberg, Germany; carmen.schwartz@zalf.de (C.S.); Joseph.Macpherson@zalf.de (J.M.);
apiorr@zalf.de (A.P.)
\* Correspondence: mostafa.shaaban@zalf.de; Tel.: +49-33432-82-481

**Abstract:** Appreciation for agricultural sustainability and ecosystem services (ESS) has received considerable attention from the scientific community. However, research has not yet systematically and sufficiently considered the spatial dimension of ESS trade-offs as a source of conflicts. Moreover, approaches for ESS management that address a wide range of beneficiaries and their interactions at landscape scale are lacking. Our main research question is how to motivate different beneficiaries of agricultural landscapes to cooperate in reducing supply–demand mismatches and accompanied conflicts, as well as to assess how different scenarios would impact relevant Sustainable Development Goals (SDGs). We present a novel and conceptual integrated model in which we employ a combination of three methodological tools: participatory geographic information system (PGIS), agent-based modelling (ABM) and a Bayesian belief network (BBN). The objective of our model simulation is to identify and manage site-specific spatial trade-off patterns and to provide decision support for shifting competitive behavior of individual stakeholders in satisfying their demand for ESS to a collective and cooperative scheme, while jointly striving to attain relevant targets outlined in the SDGs. Attached to this work is a short video depicting our conceptual model. We strongly suggest that tackling a complex social-ecological system necessitates a highly integrated modelling approach that fosters the transition from farm- to landscape-scale management, from individualistic to collective action, and from competitive to cooperative behavior.

**Keywords:** future landscapes; participatory GIS; agent-based modelling; Bayesian belief network; land use conflicts; sustainable development goals

## 1. Introduction

Agricultural land represents a pivotal, yet finite, resource for the provision of food, energy, feed, fiber, timber, water, and biodiversity [1], in addition to other socially valuable, non-marketable services for humanity [2]. It plays an important role in global development, and covers 28–37% of the Earth's surface [3]. People feel connected to landscapes where they live, work, take part in social activities, and look for relaxation. They benefit from ecosystem services (ESS) that land users and society derive from natural structures and processes [3–5]. The Intergovernmental Science-Policy Platform on Biodiversity and Ecosystem Services (IPBES) has developed a new conceptual framework presenting an alternative notion to ESS which they called "Nature's contributions to people (NCP)" [6]. The argument highlights the difference from the previous approach of the Millennium Ecosystem Assessment (MEA) through the recognition of the central and pervasive role of culture, as well as indigenous and local knowledge, in defining all links between people and nature and in understanding nature's contribution to people, respectively [6]. However, the new conceptual framework has sparked several critiques disagreeing with the justification of this paradigm shift and with the usefulness of NCP as a scientific advance [7].

Apart from this debate, the security and sustainability of supplying these benefits to humans necessitates plausible and effective decisions on land management that could be performed at different scales and cooperative levels. However, Benton [8] emphasized that ESS should be managed at the landscape scale, inasmuch as landscapes influence the ecology of farms and fields within these landscapes. This was supported by the evidence of doubling the biodiversity status of a single farm in response to the application of organic farming in neighboring farms [8–10]. Thus, farmers as a group could shape the agricultural landscape and control habitat that is important for nature conservation [11,12]. Furthermore, the flow of ESS is not only restricted to neighboring farms within a landscape [13] but also contributes to regional development via food production as well as through non-marketable socio-cultural and environmental ecosystem services, which have been shown to outweigh food production [14,15]. By managing a major share of our global landscapes through agriculture and forestry of higher or lower intensity and diversity [1], agricultural land use has a strong influence on the level at which these beneficial ESS are realized. Accordingly, a comprehensive investigation of the supply and demand of ESS, as well as the impacts of management practices, could be better performed through considering links across scales in the analysis and quantification of the off-site effects of land use management [16]. Ignoring these off-site effects could lead to sub-optimal solutions at that scale, and lead to negative feedbacks at the local scale where the initial decisions have been made [16].

Many ESS possess qualities of public goods for which public and private incentives exist to improve their provision. In effect, many policies intend to reach a balance of supply and demand of ESS, while farmers primarily respond to commodity markets [17]. However, supply and demand of ESS are locally determined [18] and there is still a lack of concepts and tools for multi-stakeholder demand-oriented ESS provision at the regional/sub-regional level, i.e., the scale at which balancing could be most effectively managed. A review of information system/decision support system (IS/DSS) tools in the field of agriculture, landscape and environmental management developed within research projects funded by the European Union (EU) during the period between 2007 and 2017 has been conducted by Zasada et al. [19]. They assessed common specifications and functionalities of knowledge transfer, including users' interaction during their development, and identified future development approaches. The study revealed insufficient involvement of stakeholders and matching of expectations. In addition, a research gap exists regarding the identification of societal demand for ESS according to varying values people actually receive from ESS [20], or utility they perceive from them [21].

Additionally, there is no clear definition of the term "demand" for ESS among a wide community of researchers. Wolff et al. [22] have stated that ESS demand can be distinguished as either the direct use/consumption of an ESS or as the desired/required level of the ESS by society [23]. Although Villamagna et al. [24] argues that only the required level of the ESS by society should be considered as ESS demand, whereas the actual consumption represents its flow [23], many studies assume consumption or the difference between the potential and actual supply as the demand [23,25–28]. In this study, we focus principally on the explicitly stated and desired level of allocated demands by different categories of stakeholders.

Indeed, it is known that conflicting demands and related differently applied land use management practices generate trade-offs among multiple ecosystem services. Vice versa, trade-offs in ESS can serve to identify varying stakeholder preferences [29,30]. However, landscape management as well as landscape design and policies have not yet sufficiently solved the challenge of managing, evaluating, and optimizing ESS trade-offs between diverging demands appropriately [29,31].

Likewise, research has not yet sufficiently considered the spatial dimension of land use conflicts in a systematical manner through approaches that allow for spatially differentiated identification and analysis of trade-offs. Among the few works focusing on spatial mismatches, Geijzendorffer et al. [32] present and apply, for assessing the match, an

analytical framework distinguishing potential supply and managed supply versus interests and expressed demands, proving the discriminative capacity for depicting stakeholder diversity, e.g., in participatory approaches and mapping of ESS. As a different possible approach, Seppelt et al. [16] point to optimization approaches (exploratory modelling) and combinations of measures. In this paper, we want to contribute to the latter, by suggesting a conceptual analytical framework towards conflict avoidance in ESS demand and supply, and by taking the following basic consideration as a starting point: In cases of regional supply gaps of single or multiple ESS, the demand of farmers' themselves, as well as stakeholders and civil society, forces them either to fulfill their demands for these services in other locations with lower conflictive conditions or, in the case of farmers, to apply appropriate land management strategies to lower gaps between potential and managed supply.

The first objective of our research is to identify hotspots of supply–demand mismatches and tradeoffs for selected ecosystem services via data collection on demands from different stakeholder groups. These mismatches will be displayed according to specific location through a participatory GIS tool. The second objective is to identify potentially resulting risks of conflicts, as well as cooperation potential between beneficiaries of ESS, and thereby tackle the spatial and temporal complexity of the social-ecological system. We approach this goal by applying a dynamic spatial agent-based modelling that combines the heterogeneity of landscapes and human behaviors. Our third objective is to assess uncertainties regarding the impact of system changes on relevant Sustainable Development Goals (SDGs) through the application of a Bayesian belief network approach. The overarching goal of this integrated model is to reduce the supply–demand gaps of ESS, to avoid conflicts between beneficiaries, to improve the sustainability of the system, and to motivate the stakeholders to consider and appreciate not only their own individual demands for ESS, but also collective global demands in their decision-making processes.

Consequently, our research questions focus on: (1) how to map the societal demands for provisional and regulatory ESS in agricultural landscapes; (2) how to evaluate the spatial supply–demand mismatches and potentially arising conflicts; (3) how actors make decisions to minimize these mismatches; (4) how their decisions would change their capitals; and (5) what are the possible consequences of the cooperative and non-cooperative interactions between actors regarding risks of land use conflicts and impacts on the SDGs?

In Section 2, we elaborate on the methodological challenges for modelling approaches and ambitions derived for this research. In Section 3, after reflection on the state of the art of existing models and tools, we develop our conceptual model based on three modules. In Section 4, we introduce two decision adaptation scenarios for demand satisfaction and exemplify it with a video. Section 5 provides a summary of our conceptual model and its future application.

## 2. Challenges of Modelling Human–Environment Interactions

Humans are characterized by different socio-demographic features and backgrounds. Human decisions with regards to land use are guided by their behavioral evaluation of possible action pathways and their preferences towards some of the evaluating aspects of these action pathways [33]. Human–environment interactions represent complex social-ecological systems [34] with an adaptive behavior. This kind of interaction behaves in a non-linear, co-evolutionary dynamic, derived from continuous adaptation and learning at multiple scales [35]. Humans and the environment are heterogeneous in nature, where individuals have their own goals and values, and therefore behave differently. Similarly, landscapes are a mosaic of many components in terms of topography, soil, vegetation, and land cover. This heterogeneity represents an essential factor of the complexity of such a system [35].

Modelling of complex social-ecological systems has received increased attention, because it is capable of representing the interactions between human demands and the environmental supply of goods and services from the available natural resources in a speci-

fied region. However, improper anthropological intervention, in our case agricultural land management, and exploitation of these natural resources, either through overconsumption or pollutant loading (e.g., pesticide, herbicide, chemical fertilizer, etc.), impairs the equilibria of ecological systems and, hence, the provisioning of ESS. This in turn has negative consequences for society as a whole, leading to conflicts and competition for resources on the local and global level as well. Although some attempts have been undertaken to manage multiple services from landscapes and tradeoffs between ESS, there is still a research gap related to the design of sophisticated modelling tools that can meet the needs of a wide range of decision-makers [2] and find the optimum solution for ESS management, and thereby to mitigate the accompanying conflicts across scales.

Integrated landscape management (ILM) promotes the concept of landscape multi-functionality of rural areas and is considered an important tool to achieve the 17 SDGs for the transition to sustainable lifestyles. ILM acknowledges the role of stakeholders in designing valuable landscapes and the transition towards green economies [5]. Although the management of ESS on the landscape-level in either heterogeneous or homogenous farms improves collective benefits, some farmers of heterogeneous farms would lose by collaborating, ending up with lowering the stability of this kind of collective action [36]. Therefore, there is additional need of a highly "uber" integrated assessment modelling approach that can demonstrate the interactive cost–benefit dynamics on the local and global scales [2,37].

### 3. A Novel Integrated Conceptual Model

Several models with different underlying methodologies have been developed for the assessment of ESS at different scales. Bagstadt et al. [38] have compared 17 decision support tools for the quantification and valuation of ESS. We briefly mention here three popular models: InVEST (Integrated Valuation of Ecosystem Services and Tradeoffs), ARIES (Artificial Intelligence for Ecosystem Services) and MIMES (Multiscale Integrated Model of Ecosystem Services).

The InVEST toolset includes models for quantifying, mapping, and valuing the benefits provided by terrestrial, freshwater, and marine systems [39]. This tool depends mainly on biophysical spatial data as model inputs and encodes ecosystem functions in deterministic models to show the outputs either in biophysical or econometric units [38]. However, they incorporate the demand in terms of existing data about beneficiaries of ESS (e.g., where people live, important cultural sites, infrastructure, etc.) [39] without an explicit statement of demand for ESS by beneficiaries. This tool is concerned mainly with the supply of ESS, which makes it a helpful tool for decision-makers in assessing the impact of changing the input parameters through management options on the provision of ESS.

The ARIES toolset has been developed based on a combination between agent-based modelling (ABM) and a Bayesian belief network (BBN), where it encodes ecosystem functions in probabilistic models [38]. It applies artificial intelligence techniques to pair locally appropriate ESS models with spatial data [40]. The ABM plays a role in quantifying the flow of ESS between ecosystems and beneficiaries in adaptive dynamics, whereas the BBN estimates the uncertainty of these flows [38]. This tool considers the demand as the consumption of ESS through quantifying the flow of these ESS. Moreover, it examines different policies to maintain the provision of ESS and tries to optimize payments for ESS (PES) in space [41].

The MIMES toolset investigates the temporal and spatial dynamics of the transformation of materials between natural, human, built, and social capitals at different scales [42]. It serves as a geospatial information system that examines tradeoffs between different ESS under various economic, policy, management and climate scenarios in space and over time [42]. However, it lacks addressing supply–demand mismatches and the instruments of handling potential conflicts arising from the interaction between different beneficiaries while being concerned with establishing an equilibrium between socio-economic benefits and the conservation of the ecosystem.

Voinov et al. [43] have assessed and categorized different methods and tools that are used for participatory modelling to assist modelers and stakeholders with their choices and decisions. They rated these methods according to their capability of producing outputs and, according to requirements for implementing participatory modelling, from low (L) to medium (M) to high (H). We selected three of these tools to complement the deficiency of one with the strength of another (see Table 1). For instance, ABM has high capability to represent the system spatially, as compared to BBN, a high capability to represent the system temporally, in quantitative and qualitative forecasts, handling uncertainty, as compared to GIS, and to support system dynamics in terms of feedback loops as compared to both BBN and GIS., whereas GIS is easier for communicating results and in modification. BBN supports ABM in quantitative and qualitative forecasts and in handling uncertainty. GIS and BBN improve the transparency of the outputs which is lacking by ABM. However, the requirements of these methods put some limitations in their application. As can be observed from Table 1, ABM requires high knowledge of the system, expertise of the modelers, to some extent consumes a lot of time, and has high costs. GIS and ABM require high computer resources as compared to BBN. The requirement of methodological expertise of stakeholders shows the lowest for GIS which supports the use of a participatory geographic information system (PGIS) in data collection.

**Table 1.** Rating of the three methods implemented in our conceptual model framework [43] (GIS = Geographic information system, ABM = Agent-based modelling, BBN = Bayesian belief network, H = high, M = medium, L = low, L/M = low-medium, M/H = medium-high).

| | Capability of Producing Outputs | | | | | | | | |
|---|---|---|---|---|---|---|---|---|---|
| | Spatial representation | Temporal representation | Qualitative forecast | Quantitative forecast | Ease of communicating results | Transparency | Ease of modification | Feedback loops supported | Handling uncertainty |
| GIS | H | L | L | L | H | M | H | L | L |
| ABM | H | H | H | H | M | L | M | H | H |
| BBN | L | L | M | M | L/M | L/M | M | L | M |

| | Requirements | | | | | |
|---|---|---|---|---|---|---|
| | Time and cost | Empirical data | Systems knowledge | Expertise of modelers | Methodological expertise of stakeholders | Computer resources |
| GIS | M | H | L/M | M | L | H |
| ABM | M/H | L/M | H | H | L/M | H |
| BBN | M/H | M | M | M/H | M | M |

In our conceptual integrated model framework, we go through four steps (see Figure 1). In the first step, we collect data about specific parameters of the social-ecological system that feed into the ABM in the next step. From the human side, we collect data, through applying the PGIS tool, about demands for ESS, perceived supply of ESS, and socio-economic demographics of the stakeholders who have been categorized based on a stakeholder analysis. We statistically analyze the demographic data of the respondents through applying factor analysis (FA), principal component analysis (PCA) and cluster analysis to analyze their behavioral differences in stating their demands for different ESS in an approach similar to that carried out by Weltin et al. [44]. From the land side, we identify the case study areas (CSAs) based on experts who conducted previous research in these areas, and we select the ESS according to a literature review showing the highly important ESS in these areas. Additionally, we use potential and status quo supply maps of ESS developed by ESS supply models existing in the literature. The following step is to feed these data into the ABM to run a simulation depicting the emergence of the social-ecological system. We run the model under different decision rules and by changing the values of some variables to test different scenarios that show the spatio-temporal evolution of supply–demand gaps,

risks of conflicts between actors, and the change in the different types of capitals of these actors. Linking the model from the local to the global scale, we assess the impacts of the system on indicators representing the relevant SDGs through applying a stakeholder workshop. In the last step, we analyze the uncertainty of the impact assessment by applying a BBN approach which ultimately gives a recommendation of which scenario would reduce supply–demand gaps, avoid conflicts and support SDGs. In the following subsections, we give more details about the three methods and how they can be applied.

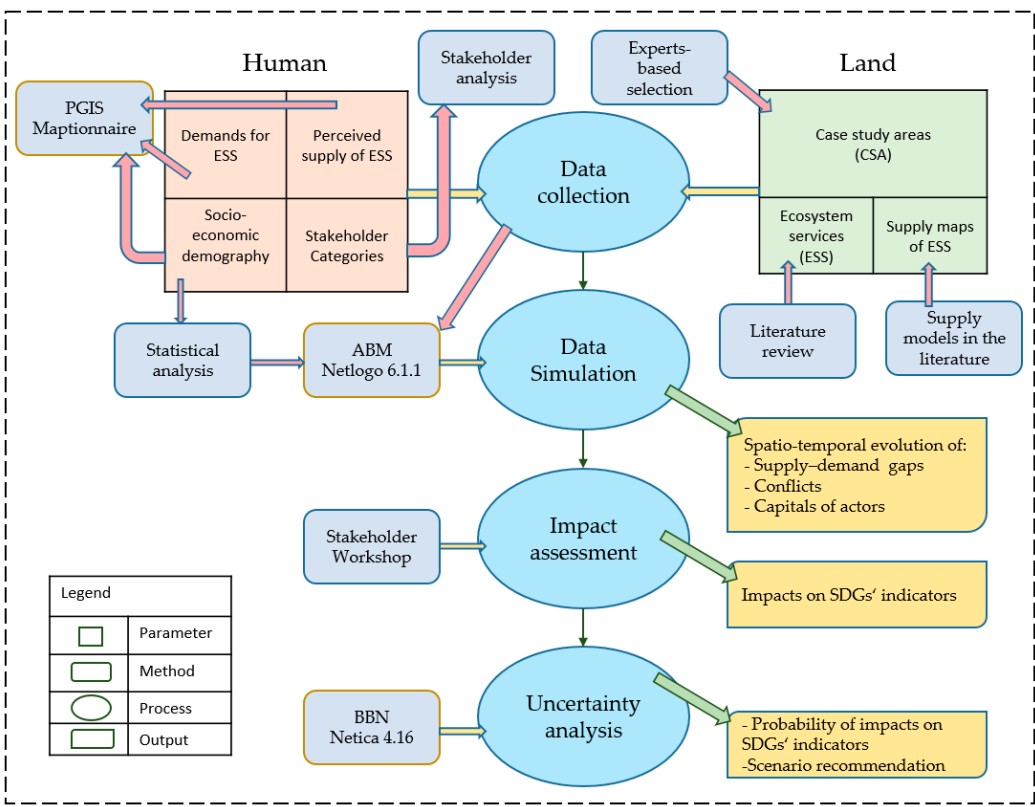

**Figure 1.** Conceptual Integrated Model Framework (PGIS = Participatory geographic information system, ABM = Agent-based modelling, BBN = Bayesian belief network, ESS = Ecosystem services, SDG = Sustainable Development Goals).

### 3.1. Participatory GIS

Different approaches have been implemented to assess the demands for ESS either through statistics, modelling, socio-economic monitoring, or interviews [45]. Most studies evaluate the demand for provisional and regulatory ESS, where demands are considered as the desired consumption rate [46] or based on biophysical data as the difference between the capacity and actual supply [28]. However, demands for cultural ESS are mostly evaluated through a GIS application survey, as has been depicted in the Social Values for Ecosystem Services (SolVES) tool developed by Sherrouse et al. [47].

Approaches using participatory geographical information systems (PGISs) help identify the demands for and the values of multiple ESS spatially through integrating all local stakeholders, thereby overcoming functionally simplified, mono-functional landscape perspectives, which have a higher land use conflict potential [48]. By combining surveys with mapping, highlighting ESS and displaying the spatial heterogeneity of ESS benefits, PGIS approaches have the advantage of considering cultural realities of communities, regions, landscapes, and ecosystems. These benefits can be derived from the appreciation of objects such as places, ecosystems and species [5].

Following the state of the art of ESS research, ecosystem functions become services only when there is a demand for these services [49]. Hence, methods for the assessment of

the demand for ecosystem services have evolved in the past years. However, only very few studies combine assessments of formulated demand with a spatial dimension; limited evidence exists on the matching of demand and supply of spatially explicit ecosystem services. Furthermore, many studies lack definition of clear beneficiaries of the ESS assessed [50]. Digital public participation GIS tools have evolved, and have been used in studies focusing on landscape planning and the assessment of land use preferences by different stakeholders. They allow for the integration of a spatial dimension in demand and preference assessments, as well as for distinguishing between different stakeholders and beneficiaries.

Translating our approach from concept to practice, we used the online map-based survey tool "Maptionnaire Community Engagement Platform" (https://maptionnaire.com/) that has the advantage of designing basic as well as map-based questionnaires, collecting data, and conveying information [51]. It helps analyze, collaborate, report, and communicate engagement projects and plans with citizens and stakeholders, and has been applied in scientific, urban and community planning projects [51]. We used it to map and assess the demands of different stakeholder groups for five ESS which were the yield of crops and biomass, soil erosion control, water availability, biodiversity, and carbon sequestration. We applied our PGIS method in four CSAs in two federal states of Germany. Three CSAs were located in Brandenburg and one in Bavaria. The four CSAs exhibited different landscape characteristics for assessing the reproducibility and robustness of our approach. The PGIS tool played an important role in our approach; we employed it to collect data about the societal demands for the selected ESS. The participants can exactly locate the areas of interest in the CSAs by drawing polygons, express their quantitative estimation and perception about the current supply of ESS, and explicitly state their demands in terms of percentage of the optimum supply in the designated area. Furthermore, we obtained additional input data regarding their decision behavior and demographics. The results of the assessment enabled the categorization and displayed varying amounts of demands for ecosystem services by different groups, and potential conflicts between stakeholders within one area became visible.

The spatial output data were analyzed in ArcGIS and then fed into the ABM software, Netlogo (see Figure 2), to be incorporated in the adaptation simulation model. A pretest survey was successfully carried out to check the quality of the questionnaire, and a larger survey was distributed. According to the feedback we received from the pretest from experts in the CSAs, we found some technical limitations of this tool. Some forms of questions lack the ability of obliging the respondents to answer them (e.g., users cannot move to the next page before answering the question), which could result in incomplete responses. Other limitations included difficulty in understanding the drawing tool and the necessity of internet availability to participate in the survey.

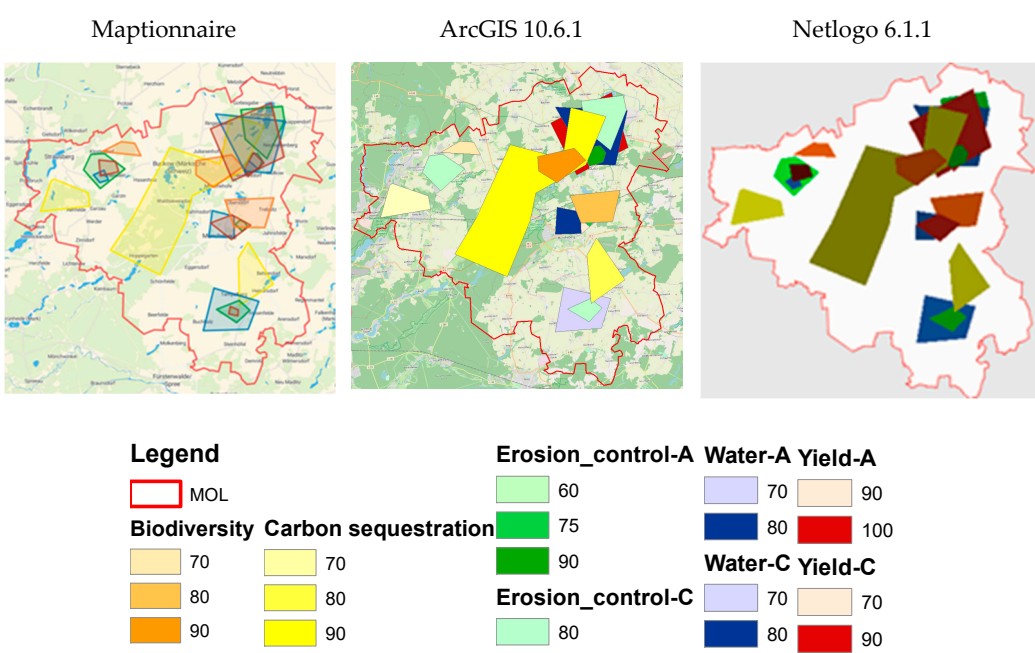

**Figure 2.** An illustrative empirical data migration from Maptionnaire to Netlogo for a case study area in Märkisch-Oderland (MOL) showing the demands for ecosystem services as a percentage of potential supply by two dummy actors A and C.

### 3.2. Agent-Based Modelling

Agent-based simulation is a bottom-up modelling approach which offers the possibility of modelling individual heterogeneity, representing explicit agent decision rules, and situating agents in geographical space [52]. It has been defined as the computational study of social agents as evolving systems of autonomous interacting agents [53]. It allows modelers to represent, in a natural way, multiple scales of analysis, the emergence of structures at the macro or societal level from individual action, as well as various kinds of adaptation and learning, which is difficult to conduct with other modelling approaches [52,54].

An agent-based model that spatially represents a complex human–environment interaction system based on heterogeneous social-ecological components and feedback mechanisms at multiple scales represents a holistic tool to assess multi-dimensional (spatial, temporal and social) tradeoffs in ESS that can be calibrated with empirical data [35]. Agent-based modelling has received increased interest for simulating human–environment interactions and their impacts on land use change and the provision of ESS in agricultural landscapes. For instance, the agricultural policy simulator (AgriPolis) model simulates the impact of agricultural policy reform on land use decisions by farmers and the concomitant impacts on ESS [12,55,56]. Similarly, Le et al. [57] developed the multi-agent system model Land-Use Dynamics Simulator (LUDAS) to simulate the spatio-temporal dynamics of a coupled human–nature system (CHNS) to assess relative impacts of policy interventions on land use changes [58]. The same model has been extended by Miyasaka et al. [35] to examine tradeoffs between ESS derived from the Sloping Land Conversion Program (SLCP), a payment for ecosystem services (PES) scheme. Sun and Müller [59] developed an integrated modelling framework that combines an ABM, BBN, and opinion dynamics models to simulate land-use decisions under the influence of PES at the household and plot levels. Pohle and Zasada [60] have developed a knowledge platform for the European project "Supporting the role of the Common agricultural policy in LAndscape valorisation: Improving the knowledge base of the contribution of landscape Management to the rural economy" (CLAIM) (http://claimknowledgeplatform.eu/claim.6.0.php) in which a combination of nine methodological tools have been applied, including an ABM and BBN, to elaborate the role of the European Common Agricultural Policy (CAP) in

landscape valorization via ESS provision and the contribution of landscape management to the rural economy. Chen et al. [61] developed the human and natural interactions under policies (HANIP) spatially explicit agent-based model to study the effectiveness of China's Natural Forest Conservation Program (NFCP) and alternative policy scenarios in a CHNS. Habib et al. [62] assessed the impacts of land use management on ESS provision using an agent-based model. Tieskens et al. [63] applied an agent-based modelling approach to explore the outcomes of future landscape change, focusing on the quality of hedgerows and cultural values in relation to scale enlargement, using a case study area renowned for its hedgerow landscapes.

Most of the existing models focus on assessing the impact of agricultural policies on the change of land use, the economic growth of the household and the concomitant impact on the provision of ESS where farmers and/or households are the main actors involved in these models. However, there is a lack of consideration of the demands for specific ESS by different categories of actors and the interaction between their decisions at the landscape scale. In general, Schulze et al. [64] have reviewed points of strengths and weaknesses of social-ecological agent-based models published in more than 100 papers according to the TRACE (TRAnsparent and Comprehensive model Evaluation) framework, which is used for documenting the development and testing of ecological models. They found a substantial gap in published models regarding their ability to imitate real systems and a lack of systematic and transparent way of model analysis for better understanding.

In this study, we applied an agent-based model that evolves from and expands on a previously developed evolutionary adaptation spatial multi-agents-based model that has been applied in other fields of research [33,65–67]). The model simulates the interaction between stakeholders from different categories encompassing farmers, foresters, retailers, inhabitants, policy makers, and others, who show demands for specific ESS in the agricultural landscapes as individual spatial agents that make decisions to fulfil their demands. These agents are differentiated by their capital and decision behavior, which is driven from their preferences of utilities. We collected different classifications of capital in the literature [68–72] and reclassified it into financial, human, social, natural, physical and cultural capital. Under each of these capitals there are multiple parameters that can be quantitatively measured and integrated into the model. The logic of the model is based on the dynamic exchange between these different types of capitals with the ultimate objective of reducing the gap between the demands for ESS and the existing supply and increasing their preferred capital. The agents prioritize the investments of their efforts according to their capacity (i.e., capital) and their preferences regarding actual and expected utilities they would receive from ESS. The major distinction of our model is that we do not focus exclusively on monetary investments and values, but also on other types of investments, such as knowledge, social activities that produce economic value as well as other non-economic values (e.g., better quality of life, happiness, improved health). This way of proceeding allows us to approach the goal of identifying decision rules and potential cooperation constellations between agents in relation to site-specific ESS.

We identified four matrixed decision rules based on the adaptive mechanism to be undertaken by the actors and the cooperation level between actors. The adaptive mechanism explains how the actors try to adapt to reach their objective. In our case, the actor has the opportunity to adapt the supply to the demand or to adapt the demand to the supply. A detailed explanation of these two decision action pathways can be found in Section 4 and in the Supplementary Material. In the cooperation level domain, we simulated non-cooperative and cooperative interactions between the actors. The non-cooperative (i.e., competitive) interaction would show the influence of other actors' decisions on reducing the values of others dividing them into winners and losers. In the cooperative interaction, actors would allocate their investments in coalitions without impairing the values of other actors, with a target of achieving a win-win situation. In real systems, a combination of all these decision rules takes place, splitting the actors into coalitions of allies and opponents, bringing in the complexity of the social-ecological system. The ABM

will allow simulating the development of the status quo of the supply–demand gap of ESS and will identify conditions, mechanisms and scenarios by which this gap can be filled and land use conflicts can be avoided. The model is currently under development and is being programmed in open access software, Netlogo 6.1.1, developed by Wilensky [73], as shown in Figure 3. It will be calibrated and tested with empirical data from the CSAs mentioned in Section 3.1.

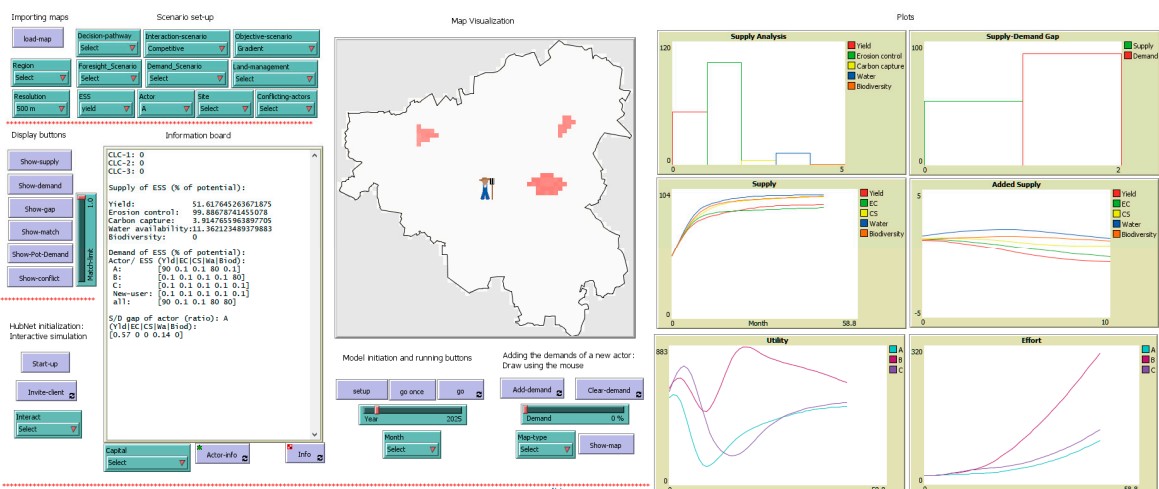

**Figure 3.** A preliminary interface of the model in Netlogo showing the supply–demand gap of yield by a dummy actor (A) in Märkisch-Oderland, Brandenburg, Germany.

### 3.3. Bayesian Belief Network

There is clear consensus that sustainable management of agricultural ESS is crucial toward attaining the United Nations Sustainable Development Goals (SDGs) [74,75]. However, the challenge remains in assessing how actions at the regional and landscape level align with achieving the global SDG targets [76]. Some studies have addressed this topic through modelling frameworks [77–80]. However, in most studies, uncertainties are rarely explicitly taken into account when assessing impacts, which is especially relevant in terms of data scarcity [81]. Here, our model attempts to bridge this gap by using a Bayesian belief network (BBN) to assess potential impacts, as well as uncertainties therein, of agents' behavior as simulated in the ABM on agricultural landscape-related SDGs. Although the 17 SDGs are all interrelated, in this study we were concerned with those goals that are more closely linked to agroforestry landscapes in terms of food (SDG2, Zero Hunger), health (SDG3, Health and Well-Being), climate (SDG13, Climate Action), ecosystems (SDG15, Life on Land), as well as consumption and production behavior (SDG12, Responsible Consumption and Production).

BBNs are graphical models of real systems that describe causal dependencies between system variables using probabilistic reasoning to make predictive and diagnostic inferences. They are specifically acknowledged for their ability to explicitly account for uncertainties in causal relationships, integrate data of different quality and accuracy, and easily update when new information is introduced to the model [82]. BBNs can be constructed based on expert knowledge, or they can be developed on empirical models and observations; however, most BBNs utilize a hybrid approach through incorporating expert knowledge and empirical data. As such, BBNs are practical in situations where empirical data is scare, or when obtaining it would be time consuming. In an established BBN, probabilities can be rapidly updated with new data, making them useful for iterative scenario analysis and adaptive management [83]. Furthermore, the ability of BBNs to integrate diverse domains of knowledge in a transparent graphical network lends itself toward modelling the complexity of socio-ecological systems and addressing value-laden questions of sus-

tainability between stakeholders [84]. Haines-Young [85] points out that BBNs facilitate deliberative–analytical processes by assisting in problem structuring and mutual learning on issues that require transdisciplinary knowledge synthesis.

For these reasons, BBNs are increasingly used as decision support tools (DSTs) to address issues in environmental management, including modelling preferences of landscape aesthetics [86], assessing trade-offs and synergies of multiple ESS according to land-use on regional scale [87], predicting crop choice of farmers based on their attitudes toward ESS [88], and quantifying the impacts of planting riparian buffer strips on flood risk and water quality [89]. Pérez-Miñana et al. [90] analyzed greenhouse gas emission potentials of farming practices using a BBN approach. Wang et al. [91] used them as DSTs for determining the effects of different irrigation strategies on crop productivity. Nash et al. [92] modeled nitrogen exports of vegetable crops using BBNs. Henriksen et al. [93] used a BBN to predict ground water quality in relation to pesticide use. Although BBNs have a rich background in agricultural and landscape management, they have not yet been used to assess how the management of agricultural landscapes contributes to attaining the SDGs.

For our model framework, experts and stakeholders will construct a BBN in a participatory modelling procedure [94], whereby the BBN will be used to validate the results of actors' decisions as simulated by the ABM (see Section 3.2) toward attaining the SDGs. Using materiality analysis [95], participants will jointly derive a set of indicators of potential impact areas (e.g., greenhouse gas emission, fertilizer and pesticide use, pollination, habitat quality, employment, livelihood, nutrition, tourism, etc.) related to the SDGs and the ABM scenarios for one specific CSA. Influence matrices [96] will be used to guide participants in developing the structure of the BBN to build impact pathways between ESS indicators and the relevant SDGs. The causal dependencies between indicators will be quantified through existing empirical measurements and expert opinion. Once implemented, the BBN will be used to conduct scenario analyses to assess trade-offs and synergies resulting from the effects of different decision pathways on ESS supply–demand and attaining the SDGs. The outcomes of the BBN will be discussed among the participants to evaluate the predictive accuracy of the models and explore potential ambiguities in perceptions. Construction of the BBN will be done using the software [97].

BBNs are static system representations; therefore, they cannot readily incorporate feedback loops, which is an important characteristic of environmental processes [83]. This can be overcome by using a dynamic BBN approach, which replicates the network and connects variables through time slices [98]. However, this method adds an extra layer of technical complexity to the modelling procedure and will remain outside the purview of this study. Additionally, BBNs that are heavily dependent on expert opinion are susceptible to problems of bias. Bias in BBNs can be introduced when variables that may bear significant influence on the system are unintentionally left out, or when experts are overconfident when estimating the strength of causal influence between variables [82]. To address this issue, the modelling procedure will include diverse range of stakeholders, accompanied by thorough discussions with participants about scenario outcomes of the BBN.

## 4. Decision Pathway Scenarios

To avoid anticipated conflicts, we apply two decision pathway scenarios through which human demands for ESS could be adapted: supply-driven demand and demand-driven supply (see Figure 4). A combination of both pathways is also possible; however, to facilitate our approach, we investigated the two approaches separately. In fact, the paradigm of these two assumption scenarios originates from economic strategies in industries and supply chain management. Dietzenbacher [99], for instance, has investigated the relationship between supply-driven and demand-driven input–output (IO) models. He stated that the demand-driven IO model (i.e., the standard Leontief model) assumes fixed input coefficients, whereas in the supply-driven model (i.e., the Ghoshian model) the output coefficients are assumed to be fixed. Amit et al. [100] have explained how the demands for a commodity are influenced by the amount of inventory displayed (supply)

on shelves of retailers. From another perspective, Motoyama and Malizia [101] formulated two hypotheses to explain start-ups. These are the demand-pull hypothesis and the supply-push hypothesis. The former argues that the amount, growth, and density of aggregate demand will stimulate start-ups in any sectors, whereas the latter argues that factors including high-tech industry concentrations, patent generation, industrial and university research activities, as well as government funding will stimulate high-tech start-ups.

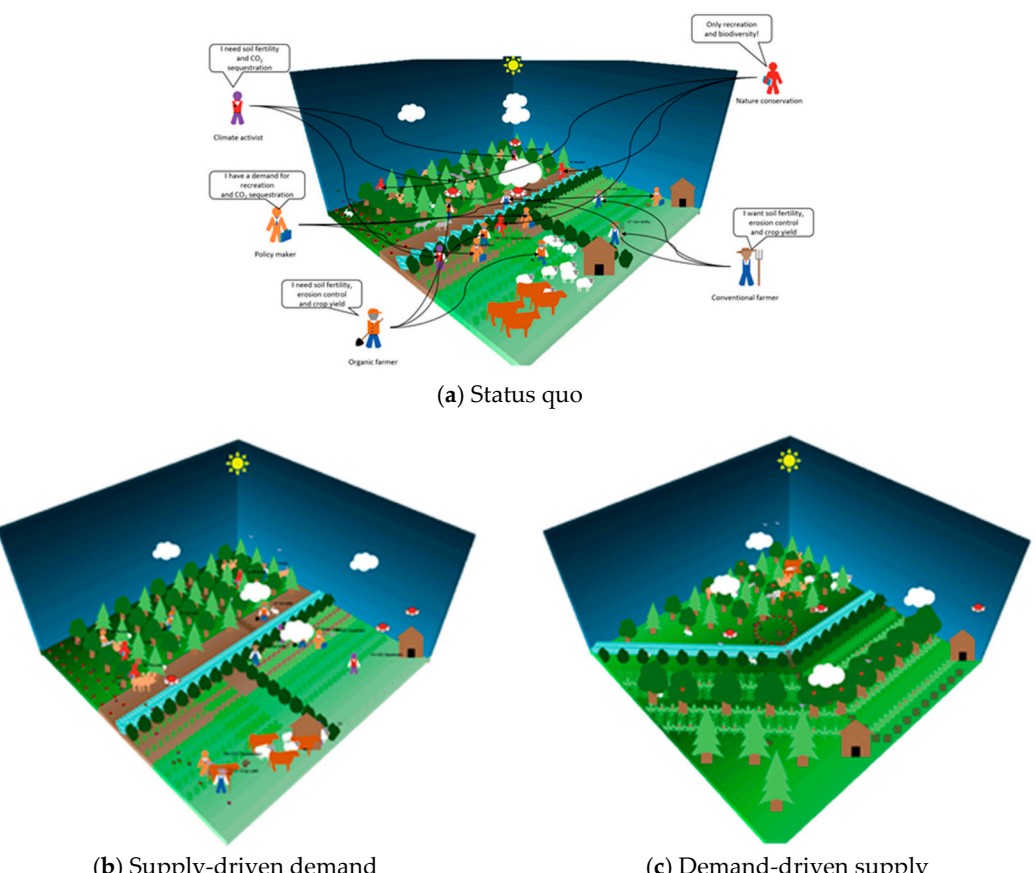

(**a**) Status quo

(**b**) Supply-driven demand　　(**c**) Demand-driven supply

**Figure 4.** Illustrative explanation of the two decision pathways: supply-driven demand and demand-driven supply. (**a**) The initial landscape before applying any decision to satisfy the demands of actors from different categories. (**b**) The supply-driven demand scenario where the actors adapt their demands in space without changing the supply. (**c**) The demand-driven supply scenario where the actors apply management options that would change the landscape in order to increase the supply and satisfy their demands.

In the supply-driven demand decision pathway (i.e., demand adapted to supply–supply push), we assume that the main driver is the result of an incapacity to influence the supply of ESS in the area where it is demanded, forcing actors to search for alternative areas or to adapt their demands to the existing level of supply. Based on Equation (1), the landscape would be divided into spatial units $k$ exhibiting different supply patterns of ESS. Hence, actors $i$ adapt the investment allocation $P$ of their efforts by comparing the marginal value of each spatial unit $v_i^{k,m}$ with the weighted average marginal value $\sum_l P_i^l v_i^l$ of all possible spatial units $l$. Actors would rank these spatial units according to the values and their preferences of the expected benefits $v_i^{k,m}$ from the demanded ESS that would be ultimately added to their capital.

$$\Delta P_i^{k,m} = \frac{\partial V_i^{k,m}}{\partial P_i^{k,m}} = \alpha_i P_i^{k,m} \left( v_i^{k,m} - \sum_l P_i^{l,n} v_i^{l,n} \right) \tag{1}$$

One example is the negotiation on compensation areas. For instance, imagine two farmers, one who owns a plot of land within a designated area (e.g., Nature Park) and wants to introduce a societally demanded crop (e.g., sunflowers) that requires higher inputs of nutrients, and another farmer outside the designated area who seeks to maintain extensive management, but wants to improve biodiversity. We assume the first farmer has to buy some equipment and has establish new market connections, which are investments with impacts that accrue over five years. In this case, the farmer compromises between maintaining management and negotiating on exchanging his/her land with the other farmer. For the nature park, a tradeoff exists between the risk of water and soil related ESS, as well as maintaining aesthetic/touristic attractiveness and direct biodiversity gains. Another example can be observed in the regional food supply and self-sufficiency paradigm. This notion refers to the ability of cities, regions and countries to obtain required alimentation within their boundaries [102]. The aim of this paradigm is to shift the food consumption behavior of consumers to short food supply chains, thus minimizing their ecological footprints.

The information needed for each involved actor is whether they are more concerned with the social values, e.g., they would prefer to keep the hotspots with high supply of recreation and touristic activities and move their demand for biodiversity also beyond the designated borders. Or, vice versa, if they are more concerned with environmental values, their preference would be to move hotspots with high biodiversity and to move flows of tourism as an extreme. In fact, the supply of these ESS are not constant, because they are affected by several factors, such as climate change. Therefore, actors would adapt their demands with these spatial changes within the constraints of their capital and through the negotiation with other beneficiaries. In this scenario, each actor could assign an individual adaptation rate $\alpha_i$ which defines how fast the actor would react to the change in the ecological system and relocate his/her demands.

In the demand-driven supply decision pathway (i.e., supply adapted to desired demand–demand pull) we assume that actors would be capable and prefer to invest their efforts in different management options $m$ instead of alternative spatial units $k$ to increase the supply of the demanded ESS in the area where they show a demand. Thus, actors would prioritize these different management options according to their capital and preferences to the features of these alternatives and could even apply a combination of management methods that could supply their demands by allocating their investments across the alternatives. This scenario predominantly applies to the option portfolio of farmers, with their decision making having the capacity to react to stated demands by adapting management in a way that favors the supply of the demanded goods. This can, for instance, refer to the currently observable change of the societal perception of farmers as not only providers of food, but also as providers of ecosystem services such as erosion control, biodiversity, and carbon sequestration in the landscape. By changing management from solely crop-focused monocultures to more diversified systems such as intercropping, agroforestry systems or the integration of perennial with annual crops, farmers can react to the changing demand by adapting supply. Regarding the previous example; both farmers could opt to manage their land independently or cooperate with other actors by establishing novel management practices and including new landscape, thereby improving biodiversity, broadening cropping patterns and increasing overall outcomes. This cooperation would be based on an agreement on the allocation of the payoff between the contributing actors.

## 5. Discussion and Conclusions

By focusing on the conceptual and methodological approach, the aim of this paper was to provide decision support for the prevention of land use conflicts from a status quo analysis of approaches and tools for the spatial coverage of trade-offs from ESS. It follows the idea that identifying and comparing supply and demand for ESS by different user groups, including farmers, will provide site-specific spatial trade-off patterns. Their analysis already allows conclusions to be drawn about cold- and hotspots for conflict

areas and involved groups. Participatory GIS for data mining and spatial analysis are recommended as an approach for this part, which we also implemented in our work. Assuming, in the near future, that digitalization in agriculture, as well as in everyday life, and an increase in citizen science activities will allow a much broader and more systematic coverage of such demand data of ESS in agricultural landscapes, a more systematic and model-based use of these data for forecasting and decision support in land management and policy will be required.

The actual knowledge gap therefore lies in the question of how spatially and demand-differentiated data obtained from the PGIS can be used for locally appropriate solutions and still allow transferability and upscaling with increasing data density. Conflict resolution approaches would build on a better available, accessible, and comprehensible knowledge of the preferences and flexibilities of different groups within the specific natural environment and its potential for ESS that may have been under-exploited to date. Such information can support negotiation processes and help to make them fair and reproducible. We introduced an agent-based model to simulate spatial allocation alternatives by suggesting user and location constellations with reduced conflict risk. We furthermore suggest cooperative management planning approaches that go beyond field and farm borders in order to spatially connect and optimize crop allocation and management towards demand-driven ESS trade-offs.

Finally, assuming frequency, explicitness, and scope of monitoring data mining to rapidly and markedly increase new paradigms for land use planning and policy would be possible. This includes spatially assigned result-based payments for ESS. Further developing our idea of demand-driven agriculture of the future and the related tools leads us to making the link to the SDGs, and hence considering concepts and models for system functioning. Here, BBN was our chosen approach for bringing primary data sources into probabilistic reasoning of relevance for impact assessment and upscaling to different governance levels. In total, the conceptual approach is set up to co-develop, evaluate, and control a data-based agricultural supply system that considers societal demand as a central and direct driver.

In this study, we only presented our conceptual model framework. This research is part of a large cooperative funding strand with midterm development ambition. After many years of experience in collaborative research, taking well thought-out complementary approaches and loose couplings of models, we now receive the opportunity for a radically different and truly integrated approach that is disruptive to current settings, such as land ownership-based instruments and incentives for land use and management. We also imply a much better public understanding of ESS in its complexity and broadness than we can assume presently. Therefore, we communicate this within our surveys, for example.

In the different parts of our research, we will soon provide methodological details and results, which are not yet provided in this paper, and doubtless will be missed here. On the other hand, due to the complexity of the approach, it will not be possible later on to present individual research papers in their embedding in the overall consideration as presented here. Above all, however, we believe that it makes sense to present our concept and the choice of methods to a scientific community for discussion at this early stage.

There are some general limitations of the approach, and specifically of the ABM, similar to any large-scale model, especially regarding the uncertainty and the probability of the outcomes. Some specific limitations are related to the use of a single initialization input data value, and based on it we assume the adaptation behavior evolution of the agents. In order to minimize this limitation, we plan to use the model in a stakeholder workshop where the actors can take part in a role-playing game and interact with each other through changing their decision actions in response to others. Another limitation relates to the supply map of ESS, where we would initially use static supply maps of ESS without the exact supply model. However, this would be upgraded in a later version of the model.

The aim of our integrated modelling approach is to provide decision support for shifting the decision behavior into the direction of satisfying the demands for ESS of all

involved actors in a landscape from one that is competitive and individualistic to one that is cooperative and collaborative, while adhering to the SDGs targets. Previous research has shown that farmers and stakeholders are aware that integrated solutions such as sustainable (intensification) strategies are context-sensitive and require common action at the regional landscape level and in local ecosystems [103]. Therefore, we are also curious to see which knowledge requirements, rules and implementation conditions practitioners consider to be particularly relevant and how we can take them into account in tool development.

**Supplementary Materials:** The following are available online at https://www.mdpi.com/2073-445 X/10/2/131/s1, Video S1: "A Conceptual Model Framework for Mapping, Analyzing and Managing Supply–Demand Mismatches of Ecosystem Services in Agricultural Landscapes".

**Author Contributions:** Authors contributed to this article according to the following: Conceptualization, all authors; methodology: Section 3.1, C.S.; Section 3.2, M.S.; Section 3.3, J.M.; software, M.S., C.S., and J.M.; resources, A.P.; writing—original draft preparation, M.S.; writing—review and editing, A.P., C.S., and J.M.; visualization, M.S.; supervision, A.P.; project administration, A.P.; funding acquisition, A.P. All authors have read and agreed to the published version of the manuscript.

**Funding:** This research was funded by the German Federal Ministry of Education and Research (BMBF) through the Digital Agriculture Knowledge and Information System (DAKIS) Project, grant number FKZ 031B0729A, and the APC was funded by the DAKIS Project.

**Institutional Review Board Statement:** Not applicable.

**Informed Consent Statement:** Not applicable.

**Data Availability Statement:** No new data were created or analyzed in this study. Data sharing is not applicable to this article.

**Acknowledgments:** We would like to thank all reviewers for their efforts and fruitful feedback on the article.

**Conflicts of Interest:** The authors declare no conflict of interest.

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
