# Peer review of "A Conceptual Model Framework for Mapping, Analyzing and Managing Supply–Demand Mismatches of Ecosystem Services in Agricultural Landscapes"

_land, doi:10.3390/land10020131_

Round 1

Reviewer 1 Report

This manuscript presents a conceptual model framework for mapping, analyzing and managing supply-demand mismatches of ecosystem services in agricultural landscapes. It proposes a reasonable concept and the choice of methods to our scientific community for discussion at this early stage. It is suitable for publication in the journal.

Author Response

Dear Reviewer,

Many thanks for your valuable comments.

Best Regards,

The authors

Reviewer 2 Report

The manuscript covers an interesting topic. It is well written, while methodology and results are well presented and documented. 

However, there a few minor correction that need to be considered prior to its final acceptance.

Abbreviations should be mentioned at each first appearance in the manuscript.

Reference citations within the manuscript need attention. For example:

Line 160: consider change to Bagstadt et al. [38]

Line 188: consider change to Voinov et al. [43] 

Figure 1 needs attention (see middle column lines 232-5, where text is not clearly presented)

Check also manuscript formatting to adopt Land template.

Author Response

Dear Reviewer,

We would like to thank you very much for your fruitful comments and for making us aware of many important issues in our paper with the aim of improving its quality. We appreciate all your comments and to the best of our knowledge we addressed your comments as follows:

1. Abbreviations should be mentioned at each first appearance in the manuscript.

We thank the reviewer for making us aware of this. We considered this point in our revised version.

2. Reference citations within the manuscript need attention. For example:

Line 160: consider change to Bagstadt et al. [38]

Line 188: consider change to Voinov et al. [43]

We thank the reviewer for this notice. We rechecked and modified all in-text references.

3. Figure 1 needs attention (see middle column lines 232-5, where text is not clearly presented)

We thank the reviewer for this notice. We modified the figure to make it simpler and to avoid misalignment of the text.

4. Check also manuscript formatting to adopt Land template.

We appreciate your comment. We checked the manuscript formatting and matched it with Land template.

Best Regards,

The authors

Reviewer 3 Report

This is an interesting article which deals with how to collaboratively achieve management of agricultural landscapes for optimum ecosystem service benefit.  The article proposes a conceptual model that combines three spatial methods to assist in decision support for landscape management for ecosystem services enabling the the opinions of relevant different stakeholders to be considered. 

The article should be interest to Land readers however it is current form there are many English language errors that need to be corrected.  I also found the flow of the article difficult to follow in places. 

To improve the article I feel that more clarity could be given by more explanation directly linking back to figure 1 and table 1.    There was no real in text direct explanation of these.  (There was also a misalignment of the text in figure 1 that needs to be amended.). Figure 1 is particularly important as this outlines the conceptual model framework so greater explanation of how the integration of the model would be exceptionally useful.   It was not clear from the figure or text how the 3 methods would be used. 

Table 1 also needs greater explanation.  There was no key to this table.  It can be assumed that H is high, L is low etc but this should be added.  

I was also confused by the section that describes the application of the PGIS to the CSAs as the description jumps from conceptual to application.  There is later discussion around the fact that there will be application of the model in subsequent papers so the authors need to consider  how much application is included in this paper.  

At line 277 the authors mention the software maptionnaire with no references and description of this - some extra information here would be useful.

I also felt that there could be more clarity around the aims and goals of this paper as these tended to vary depending on which section of the paper they were discussed. 

At the end of the introduction chapters were referred to - these should be referred to as sections.  

Author Response

Dear Reviewer,

We would like to thank you very much for your fruitful comments and for making us aware of many important issues in our paper with the aim of improving its quality. We appreciate all your comments and to the best of our knowledge we addressed your comments as follows:

1. In its current form there are many English language errors that need to be corrected.

We appreciate your comment. One of the coauthors who is a native English speaker rechecked it.

2. To improve the article I feel that more clarity could be given by more explanation directly linking back to figure 1 and table 1. There was no real in text direct explanation of these.

We thank the reviewer for making us aware of this. We gave more explanation about figure 1 (L212 – 232) and table 1 (L193 – 208) and made a link between them.

3. There was also a misalignment of the text in figure 1 that needs to be amended.

We thank the reviewer for this notice. We modified the figure to make it easier to understand. To avoid misalignment of the text, we developed it in PowerPoint and included it as an image.

4. Figure 1 is particularly important as this outlines the conceptual model framework so greater explanation of how the integration of the model would be exceptionally useful.

We thank the author for making us aware of this. We give more explanation about the integration of the model (L212 – 232).

5. It was not clear from the figure or text how the 3 methods would be used.

We appreciate your comment. We gave more explanation about the three methods in section 3 and how they would be used and we modified the figure to make it clearer.

6. Table 1 also needs greater explanation. There was no key to this table.  It can be assumed that H is high, L is low etc but this should be added. 

We thank the reviewer for making us aware of this. We gave more explanation to the table (L193 – 208) and explained the abbreviations in the caption of the table.

7. I was also confused by the section that describes the application of the PGIS to the CSAs as the description jumps from conceptual to application. There is later discussion around the fact that there will be application of the model in subsequent papers so the authors need to consider how much application is included in this paper. 

We appreciate your comment. This article presents mainly our conceptual approach however, we wanted to shed the light on the applicability of the model that we develop and apply in practice so that the reader do not perceive it as only a theoretical idea. We added a sentence explaining this transition from concept to practice in L262.

8. At line 277 the authors mention the software maptionnaire with no references and description of this - some extra information here would be useful.

We thank the reviewer for making us aware of this. We added a reference and a brief description of this tool in the text (L262 – 266).

9. I also felt that there could be more clarity around the aims and goals of this paper as these tended to vary depending on which section of the paper they were discussed.

We appreciate your comment. We clarified and specified the aims and goals of the paper in the abstract L22, in the introduction L108 – 120, in the discussion and conclusions L516 and L569.

10. At the end of the introduction chapters were referred to - these should be referred to as sections.

We thank the reviewer for making us aware of this mistake. We changed all mistakenly written chapters to sections.

Best Regards,

The authors

Round 2

Reviewer 2 Report

The revised version of the manuscript adopted all suggested comments. Thus, it is now suitable for publication.

Author Response

Dear Reviewer,

many thanks for your valuable and fruitfull comments that helped us to improve the manuscript.

Best Regards,

Mostafa Shaaban

Reviewer 3 Report

The paper is much improved by the revisions made.

There are just a few minor changes to be made

Line 49 - remove 'a' before plausible

Line 64 - add 'to' after lead

Line 110 - add 'to' before according

Line 154 -'mitigating' should probably be 'mitigate'

Line 189 - comma needed after However

Line 242 - not sure about the use of 'On the contrary' here - might be better to say in a different way

Line 539 - sentence not clear

Line 576 - comma needed after Therefore

Author Response

Dear Reviewer,

we would like to thank you for making us aware of these points.

We addressed all of them and kept the "track changes" funtion on so that you could check them easily. 

Best Regards,

Mostafa Shaaban